# Gender Identity: The Human Right of Depathologization

**DOI:** 10.3390/ijerph16060978

**Published:** 2019-03-18

**Authors:** Maria Elisa Castro-Peraza, Jesús Manuel García-Acosta, Naira Delgado, Ana María Perdomo-Hernández, Maria Inmaculada Sosa-Alvarez, Rosa Llabrés-Solé, Nieves Doria Lorenzo-Rocha

**Affiliations:** 1Faculty of Nursing N. S. Candelaria, University of La Laguna, Canary Islands Public Health Service, 38010 Tenerife, Canary Islands, Spain; mcastrop@ull.edu.es (M.E.C.-P.); extaperdomo@ull.edu.es (A.M.P.-H.); isalvar@ull.edu.es (M.I.S.-A.); rosallabres@telefonica.net (R.L.-S.); extnlorenzo@ull.edu.es (N.D.L.-R.); 2Faculty of Psychology, University of La Laguna, 38071 Tenerife, Canary Islands, Spain; ndelgado@ull.edu.es

**Keywords:** trans, transgender, gender identity, human rights, right to health, non-discrimination, pathologization

## Abstract

*Background*: Transgender people have a gender identity different from the one allocated to them at birth. In many countries, transsexualism and transgenderism are considered mental illnesses under the diagnosis of gender dysphoria. This pathologization impacts on human rights. *Main content*: The United Nations (UN) has denounced violations against trans-people, including attacks, forced medical treatments, lack of legal gender recognition, and discrimination in the areas of education, employment, access to healthcare, and justice. The UN has linked these violations directly with discriminatory diagnostic classifications that pathologize gender diversity. Trans-people have been pathologized by psycho-medical classification and laws all around the world, with a different impact depending on countries. This paper argues that pathologization infringes infringes upon a wide range of human rights such as; civil, economic, social cultural and also the access to medical care. *Conclusions*: The current situation for trans-people with respect to legal healthcare matters, depends on the country. Human rights are universal, not a question for cultural interpretation. They are the minimum that every human being must have assured only by the fact of being human. Countries must protect these rights by regulating trans-pathologization with special attention dedicated to intersex people and their specific needs.

## 1. Introduction

The sex that we are officially assigned at birth (male or female) is based upon our physical features. This might not, however, match our gender identity—that is, the way we feel and think about our gender. A trans person is someone who identifies with a different gender and/or expresses their gender identity differently from the gender assigned at birth [1]. 

To clarify, in the framework of definitions, the follow must be consider: ‘*gender identity*’: each person’s deeply felt internal and individual experience of gender, which may or may not correspond with the sex assigned at birth, including the personal sense of the body (which may involve, if freely chosen, modification of bodily appearance or function by medical, surgical or other means) and other expressions of gender, including dress, speech and mannerisms [2]; ‘*gender expression*’: each person’s presentation of the person’s gender through physical appearance—including dress, hairstyles, accessories, cosmetics—and mannerisms, speech, behavioral patterns, names and personal references, and noting further that gender expression may or may not conform to a person’s gender identity [3]; ‘*sex characteristics*’: each person’s physical features relating to sex, including genitalia and other sexual and reproductive anatomy, chromosomes, hormones, and secondary physical features emerging from puberty [3].

In the context of this paper, the term ‘*trans*’ includes people whose gender identities differ from their allocated sex at birth, both female-to-male/FtM and male-to-female/MtF trans persons, and genderqueer (pangender, fluid or otherwise) people who identify beyond binary sex models [4]. Trans experiences around the world are diverse but have something in common: being trans is still considered a pathology [5]. ‘*Pathologization*’ can be defined as the psycho-medical, legal, and cultural practice of identifying a feature, an individual, or a population as intrinsically disordered. Trans-people are defined as inherently pathological [5]. The consequences of pathologization include human rights violations, which impacts the access to healthcare to trans people, the right to corporal integrity, and limit the right to a legal personality. All these definitions are necessary to understand the timeline of depathologization movements and their implications on change.

The aim of this paper is to show the human rights implications of pathologization of gender identity, the impact on healthcare, as well as the need of depathologizing trans people.

## 2. Depathologization Perspectives

The Universal Declaration of Human Rights (Paris, December 1948) states the following: “All human beings are born free and equal in dignity and rights. They are endowed with reason and conscience and should act towards one another in a spirit of brotherhood. Everyone is entitled to all the rights and freedoms set forth in this Declaration, without distinction of any kind, such as race, colour, sex, language, religion, political or other opinion, national or social origin, property, birth or other status” [6]. The first article of the constitution of the World Health Organization (WHO) states: “The objectives of WHO… shall be the attainment by all peoples of the highest possible level of health” [7]. Thus, all human rights, including the right of recognition before the law, the right to the highest attainable standard of health, the right to body integrity, the right to found a family, and the right to be free from degrading treatment, among others, apply equally to all human beings, including those who may be trans or gender diverse.

In the last 10 years, there have been movements in favour of the right of depathologization, based on the right to health and to non-discrimination. With the opportunity of reviewing the ICD and DSM manuals, an international movement for trans depathologization has emerged. This movement demands, among other things, the withdrawal of classification as a mental disorder, of gender transition processes. However, there is a very important issue that is discovered with the action of this movement: the ‘democratization’ in the process of depathologization. Currently, proposals for depathologization in health care, human rights frameworks and processes of legal recognition of gender are changing towards a ‘democratized turn’ [8].

It has been a lengthy process to reach the achievements of present day, where different diversities have been depathologized. Firstly, sexual diversity, later, diversity of gender, in which we are currently immersed and most recently, body diversity with the aid of the intersex movements. 

## 3. Historical Precedent: Depathologization of Homosexuality or Sexual Diversity

Instances of criminalization, discrimination, and homophobic violence have occurred throughout the twentieth century. Homosexuality was pathologized as a mental disorder and the application of reparative therapies, aimed at convincing the person to adopt a heterosexual sexuality, was common [9]. From the 1970s to the 1990s, processes of depathologization and protection of rights occurred. Lesbian, gay, and bisexual movements arose. The diagnostic classification of homosexuality was removed from international diseases classifications, and homosexuality was included in an international human rights framework. Concurrently, social and legislative advances were made [9] but not equally distributed worldwide.

## 4. Current Moment: Depathologization of Transgenderism or Gender Diversity

### 4.1. Depathologization as a Healthcare Issue 

In the 1980s, 1990s, and 2000s there were protests against the diagnostic of transsexualism as a mental disorder, a classification that led to pathologization. A critical review of the healthcare model of attention was initiated. Since 2007, coordinated manifestations for trans-depathologization have taken place and activism has occurred on an international scale [10]. Critical challenges by trans people about their healthcare have often evolved through health social movements. In recent years, international activism has been prominent with the Stop Trans Pathologization (STP) campaign (beginning in 2007). The main objectives of STP are the removal of the classification of gender transition processes as a mental disorder from diagnostic manuals, access to state-funded trans healthcare, change of trans healthcare towards an informed consent approach model, legal gender recognition without medical requirements, depathologization of gender diversity in childhood, and protection from transphobic violence [11]. In addition to the STP, GATE [12], ILGA [13] and TGEU [14] stand out. These networks of international activists are succeeding in influencing the policies of countries and organizations such as the WHO or the UN. The demands for depathologization are also supported by European bodies such as the Council of Europe and the European Parliament [15] as well as professional associations such as the WPATH [16]. The case of WPATH is of particular interest because they combine the activism with ‘the promotion of the highest standards of health care for Transsexual, Transgender, and Gender Nonconforming People’ [17]. In that way, the SOC-7 (Standards Of Care, version 7) of the WPATH contains several improvements: the conceptualization of gender transitions as non-pathological, the use of non-discriminatory language, the inclusion of broad expressions of gender, transitions and identities, the need to adapt and make more flexible the trans healthcare pathways, the explicit condemnation of reparative therapies and the recognition of the cultural diversity of trans people [17]. Simultaneously, activist networks criticize the use of a pathologizing diagnostic framework and the use of a process model that moves away from the current model of person-centered care [8,17].

As mentioned above, the contribution of the activist movements has been able to contribute to a ‘democratized turn’ in the process and revision of depathologization proposals to be included in the ICD and the DSM. However, a recent review shows that these movements for depathologization have little impact on the clinical practices of trans-healthcare. But a change towards models of informed consent and person-centered care is being achieves, even in limited health interventions; but in a progressive and growing way [8]. An informed accompaniment and sharing decision-making model is still necessary instead of just an evaluation model. That is, a model of trans-specific healthcare.

### 4.2. Evolution of the Diagnostic Classification in the ICD and the DSM

The International Classification Of Diseases (ICD-10) [18], a standard diagnostic tool published by WHO, lists “Transsexualism” and other “gender identity disorders” in the chapter of “Mental and behavioral disorders”. The American Psychiatric Association publishes the Diagnostic and Statistical Manual of Mental Disorders (DSM-5) [19] in which the concept is termed “gender dysphoria”. Despite the name change, both are considered mental illnesses, since they are included in this list. Homosexuality was removed from the DSM in 1973 and from the ICD in 1975. 

In the newest edition of the ICD, ICD-11 [20], which was published in June 2018 and will be presented for approval at the World Health Assembly in 2019, trans identities have finally been removed from the mental health chapter. However, intersex people are still being pathologized and called disordered. (Table 1).

Defining gender diversity as an illness or otherwise abnormal is unfounded, discriminatory, and without demonstrable clinical utility. Psychological trauma and suffering are not inherent to trans-people but are the result of society’s failure to embrace body diversity. This has been partially solved in the ICD-11, which now considers “gender incongruence” as a condition instead of an illness. There have been advances in the legal and medical fields regarding the conceptualization of sexual, body, and gender diversity, resulting in a shift from considering it a mental disorder to recognizing it as a human right. But there are still actions to take. In the ICD-11 the new condition is “gender incongruence”, it would be pertinent to reflect on what the word “incongruence” means. When does the transsexual-transgender person stop being incongruous with his/her gender? Do they ever stop? When does it begin? Finishing hormonal treatment? When completing surgery?

Activism has argued for the withdrawal of the diagnostic classification of gender diversity in childhood. Children are very vulnerable to the dynamics of pathologization, institutional violence and non-consensual treatments. Given that fact, it is necessary to argue that the diagnosis of “gender dysphoria” in children in DSM-V or “gender incongruence” in childhood in ICD-11 has no clinical utility because children do not need treatment (until there are close to puberty). However, psychological support services, if needed, do not require a diagnosis.

### 4.3. Depathologization as a Human Right Issue

Key human rights mechanisms of the United Nations (UN) have affirmed the States members’ obligation to ensure the effective protection of all persons from discrimination based on sexual orientation or gender identity. However, the international response has been fragmented and inconsistent, creating the need for a consistent understanding of the comprehensive regime of international human rights law and its application to issues of sexual orientation and gender identity [22]. In this context of international human rights law, the depathologization of transsexuality has been discussed in a limited way. Thus, depathologization can be included in Article 8 of the European Convention on Human Rights (ECHR) that ensures the right to respect for privacy [23]. Gender identity is one of the most intimate aspects of the person. Article 8 protects the right to development and personal identity as well as the physical and psychological integrity of the person. Being labeled with a mental illness threatens the integrity of the person (according to article 8). Depathologization is based on the right to health and non-discrimination, particularly because of the stigma associated with mental illness and how this affects trans people [8,24]. At present, a growing body of norms and laws tries to support these views, an important example being the Yogyakarta Principles [22].

The principles of Yogyakarta are not legally binding but have been widely recognized internationally as an important tool for member states to identify, respect and protect the human rights of all people regardless of their sexual orientation or gender identity [22]. (Table 2).

Despite all of this, discrimination and transphobic violence still exist [25]. Each year, the Transrespect-versus-Transphobia Worldwide (TvT) team publishes the figures of hate crimes against gender-diverse people [26]. The 2018 update informed a total of 369 cases of reported killings between October 1, 2017, and September 30, 2018, constituting an increase of 44 cases compared with the previous year, for a total of 2982 reported cases in 72 countries worldwide since 2008 [26].

### 4.4. Depathologization as a Legal Issue

With respect to the legal recognition of gender, activism towards trans depathologization demands that future gender identity laws do not contemplate medical requirements or restrictions related to marital status, age or nationality [8]. In fact, activism insists that these requirements need to be removed from existing laws [15]. Activists for depathologization consider that the legal requirements needed for hormonal therapy, sterilization, genital surgery or divorce constitutes a violation of human rights and an attack on the physical integrity and the right to a family life. In addition, the activists emphasize that the medical requirements are contrary to the rights established in the Universal Declaration of Human Rights and the principles of Yogyakarta [8].

In recent years, a growing number of countries have modified their legislation to accommodate non-binary classifications of sex and gender. According to TGEU, 34 countries in Europe require a mental health diagnosis in legal gender recognition (Figure 1). The TvT published the current situation for legal gender recognition [26] (Table 3).

In recent years, countries worldwide are increasingly expanding male/female binary sex classifications to recognize a third status. This status should include intersex people and, broadly, all non-binary sex and gender classifications. A geographically diverse range of countries, such as Australia, Austria, Bangladesh, Canada, Malta, Nepal, New Zealand, and Pakistan allow official registration under a third sex in addition to male and female; treating sex not through biology but by criteria of gender [26,27].

## 5. Uncertain Future: Depathologization of Intersexuality or Body Diversity

Intersex people are born with physical, hormonal or genetic features that are neither wholly female nor wholly male; a combination of female and male; or neither female nor male. Intersex comprises a range of physical traits or variations that lie between ideals of male and female [28]. With some exceptions, intersex bodies are generally healthy. Infants, children, and adolescents born with intersex bodies are often subjected to medical interventions to “normalize” sex characteristics, which are not based on evidence, but on clinical feelings, beliefs, and narrow social norms [29]. However, there are doubts as to whether these interventions (surgeries) are adequate. Research conducted has shown that early surgery to shape the genitals of intersex children, accompanied by a socialization within the assigned gender, results in a better or more “typical” childhood. On the contrary, these surgeries usually result in psychological difficulties that affect the child and the adult that the child will become [27]. The consequences of interventions and associated examinations include the need for lifelong hormone replacement, repeat surgeries, lack of sexual function and sensation, incorrect gender assignment, and trauma [29].

In recent years, an international intersex movement has emerged due to human rights violations suffered by intersex people, such as genital surgeries and other non-consensual treatments. Activists are demanding a series of measures such as the cessation of non-consensual surgical treatments, depathologization in diagnostic classifications, creation of the category of “third gender” -but only if it is open to all persons, abolition of the civil registry of sex, cessation of practices of stigmatization, and reparation of iatrogenic damage [30].

On 14 February 2019, the European Parliament adopted a landmark resolution on the rights of intersex people [31]. By adopting this resolution, the European Parliament establishes a clear standard within the European Union for the protection of the physical integrity and the human rights of intersex people. The resolution complements the pioneering intersex resolution of 2017: “Promoting human rights and eliminating discrimination against intersex people”, approved by the Parliamentary Assembly of the Council of Europe. In this resolution, the European Parliament “strongly condemns the normalizing treatments of sex and surgical interventions” and urges the Member States to adopt legislation that urgently protects the bodily integrity of intersex people. It also confirms that intersex people are exposed to numerous types of violence and discrimination in the European Union and asks the European Commissions and Member States to propose legislation to address these problems. Other issues addressed by the resolution include the need for adequate counselling and support for intersex people and their families, measures to end the stigma and pathologization faced by intersex people and increased funding for civil society organizations led by intersex people [31]. Sadly, intersex people have been excluded from the process of depathologization and are included in the ICD-11 as “malformative disorders of sex development”. There are still multiple key changes that need to be made

## 6. Conclusions

Historically, transgender and gender diverse people have been faced with social and legal barriers towards their freedom and dignity. United Nations international human rights principles provide protection for people on the basis of sexual orientation, gender identity and intersex status [28]. Over the last decade, international activism has emerged for trans and intersex depathologization with the historical antecedent of the depathologization of homosexuality.

The primary reason for the discrimination and poor access to healthcare of trans-people lies in the stigma associated with mental illness. There is an obvious connection between pathologization and the difficulties to obtain a change in name or legal gender because a psychiatric diagnosis is needed [24]. It seems contradictory that for a civil, legal, and administrative matter there must be a medical diagnosis.

Defining gender diversity as an illness or otherwise abnormal is unfounded, discriminatory, and without demonstrable clinical utility. The fact that only gender diverse people are pathologized constitutes unequal treatment, resulting in a violation of the right to non-discrimination.

Another major concern is childhood pathologization. This diagnosis of “gender dysphoria” in children in DSM-V or “gender incongruence” in childhood in ICD-11 has no clinical utility because children do not need treatment.

Countries have certain obligations such as to not discriminate and to protect the right to healthcare of their citizens. Thus, trans depathologization as a human right implies the Right to the Highest Attainable Standard of Health; that is access to trans-specific healthcare. So, the right to access trans-specific healthcare serves as a justification to claim the right to depathologization.

## Figures and Tables

**Figure 1 ijerph-16-00978-f001:**
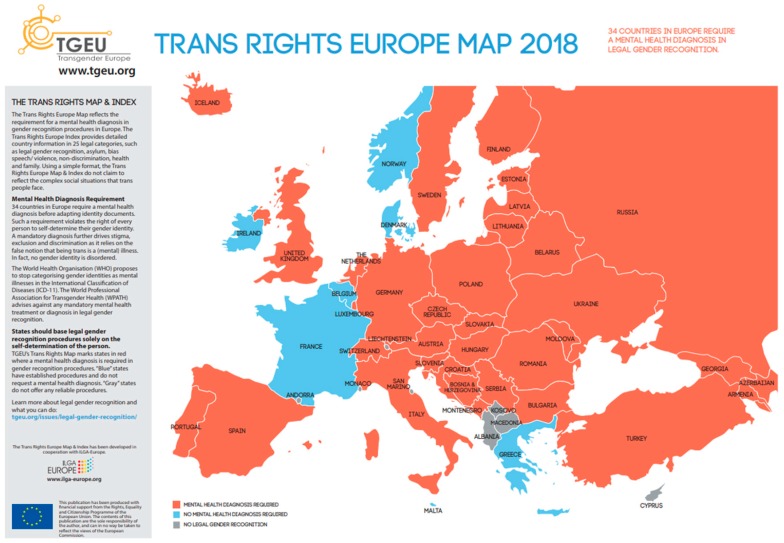
Trans Rights Europe Map (source tgeu.org).

**Table 1 ijerph-16-00978-t001:** ICD and DSM.

**DSM IV-TR** (2001) [21] DSM IV (1994)	11. Sexual and gender identity disorders	Sexual dysfunctions Paraphilias Gender Identity Disorders (GID)	F.64.2 GID in children F.64.0 GID in adolescents or adults F.64.9 GID not otherwise specified (intersex, cross-dressing behaviour)
**DSM V** (2013) [19]		Sexual Dysfunctions Gender Dysphoria (GD)	F.64.2 GD in children F.64.0 GD in adolescents or adults F.64.8 GD not otherwise specified
**ICD 10** (2016) [18]	5. Mental and behavioural disorders	Disorders of adult personality and behaviour	F.64 Gender identity disorders F.66 Psychological and behavioural disorders associated with sexual development and orientation
**ICD 11** (2018) [20]	17. Conditions related to sexual health	Sexual dysfunctions Gender incongruence (GI)	HA60. GI of adolescence and adulthood HA61. GI in childhood 5A71. Adrogenital disorders
	20 Developmental anomalies	Malformative disorders of sex development	pseudohermaphroditism
	24. Factors influencing health status and contact with health services		Gender incongruence

**Table 2 ijerph-16-00978-t002:** Yogyakarta Principles. Extract.

Principle 3: The Right to recognition before the law (without requirements such as hormone therapy, sterilization or surgery. All of these infringe upon human rights *) Principle 17: The Right to the Highest Attainable Standard of Health Principle 18: Protection from Medical Abuses Principle 31 (YP+10): The Right to Legal Recognition Principle 32 (YP+10): The Right to Bodily and Mental Integrity Principio 37 (YP+10): The Right to Truth

(*) note from the author.

**Table 3 ijerph-16-00978-t003:** The current worldwide situation for legal gender recognition (extracting from TvT).

Country	Change of Name				Change of Gender					
	Yes	Pathol. Requirement	Sterilization /Surgery Requirement	No	Yes	Pathol. Requirement	Sterilization/Surgery Requirement	No	Keeping Marriage Possible /Divorce necessary	More Than Two Gender Option
**Argentina (2012) ***	x				x				No data	two
**Australia**	x				x				marriage	three
**Belgium**		x	x			x	x		divorce	two
**Bulgaria**		x				x			divorce	two
**Botswana**	x (different)				x (different)		x		No data	two
**Chile**				x				x		
**China**		x	x			x	x		divorce	two
**Colombia**		x				x	x		divorce	two
**Cuba**		x	x			x	x		divorce	two
**Germany**		x				x			divorce	two
**Denmark (2014) ***	x				x				marriage	two
**Egypt**				x				x		
**Georgia**	x					x	x		divorce	two
**Greece**		x	x			x	x		divorce	two
**Spain**		x	x			x	x		marriage	two
**Finland**	x					x	x		divorce	two
**France**		x	x			x	x		divorce	two
**Croatia**	x					x			divorce	two
**India**	x				x				divorce	three
**Iceland**		x	x			x	x		marriage	two
**Ireland (2015)***	x				x					two
**Italy**		x	x			x	x		divorce	two
**Japan**	x					x	x		divorce	Two
**Kenia**				x				x		
**South Korea**	x					x	x		No data	two
**Malta (2015)***	x				x				marriage	three
**Mexico**		x	x			x	x		marriage	two
**Netherlands**	x					x			marriage	two
**Norway**	x					x	x		marriage	two
**Nepal**				x	x				No data	three
**New Zealand**	x					x			marriage	three
**Portugal**		x				x			marriage	two
**Romania**		x				x	x		divorce	two
**Russia**		x				x			marriage	two
**Sweden**	x				x				marriage	two
**Singapore**		x	x			x	x		divorce	two
**Switzerland**		x	x			x	x		marriage	two
**United Kingdom**	x					x			divorce	two
**United States**	x				x	x (some parts)	x (some parts)		marriage	two
**Venezuela**	x							x		
**South Africa**	x					x			No data	two
**Taiwan**	x					x	x		marriage	two
**Turkey**				x		x	x		divorce	two

(*) Gender Recognition laws without medical requirements, approved in the aforementioned year.

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
