# Peer review of "Gender Identity: The Human Right of Depathologization"

_ijerph, 2019, doi:10.3390/ijerph16060978_

Round 1
Reviewer 1 Report
Title: Highly engaging. Many would read this.
Abstract: Change the definition of transgender, as it relies on cisnormative pronouns (his/her) and the wording is a little garbled. Make it person first and remember to all for non-binary gender models (dominant in youth in many contexts). Try: ‘Transgender people have a gender identity different from the one allocated to them at birth’.
Introduction: Again, the definition is a bit old school and institutional given the radical aim of the piece. Please refer to the literature. Show knowledge of and refer to both:
*the much broader definitions offering consideration of both binary and non-binary trans identities used in Smith et al. (2014. From Blues to Rainbows. ARCSHS: Melbourne and
*the decolonising work on trans identities (showing trans as historically and currently including many cultural conceptions) in the Southern theory paper: Jones, T. (2018). Trump, trans students and transnational progress. Sex Education 18(4). 479-494.
Depathologisation Perspectives: The authors should argue a bit more for what is original here and spend less of the discussion space re-hashing existing data. The large table, whilst interesting, appears to come from elsewhere? Spend more time on the rights analyses.
Conclusion: this section is too long and includes new information (on intersex)... cut the section down and use it to expand on the relevance of points not add new things. Put those ideas earlier.
Author Response
Point 1:
Abstract: Change the definition of transgender, as it relies on cisnormative pronouns (his/her) and the wording is a little garbled. Make it person first and remember to all for non-binary gender models (dominant in youth in many contexts). Try: ‘Transgender people have a gender identity different from the one allocated to them at birt
Response 1:
Thank you very much for your input. It's already changed. I find the reason you offer very interesting: 'make it first person and remember to all for non-binary gender models ...'. Also, the use of he/she or his / her makes the text harder to read. In fact, in my native language (Spanish) we have a neutral option very suitable for these occasions. Studying English, I have seen the possibility of using they / their. A more current language, more inclusive and that segregates less ... it seems very successful
Point 2:
Introduction: Again, the definition is a bit old school and institutional given the radical aim of the piece. Please refer to the literature. Show knowledge of and refer to both:
*the much broader definitions offering consideration of both binary and non-binary trans identities used in Smith et al. (2014. From Blues to Rainbows. ARCSHS: Melbourne and
*the decolonising work on trans identities (showing trans as historically and currently including many cultural conceptions) in the Southern theory paper: Jones, T. (2018). Trump, trans students and transnational progress. Sex Education 18(4). 479-494
Response 2:
Thank you very much for your comment. It has never been our intention to appear antiquated or overly academic. In any case, we agree with you, that our paper improves with the contributions of the two texts that you suggest. They are already included, even in several places of our document
Point 3:
Depathologisation Perspectives: The authors should argue a bit more for what is original here and spend less of the discussion space re-hashing existing data. The large table, whilst interesting, appears to come from elsewhere? Spend more time on the rights analyses
Response 3:
Thank you very much because you have made us reflect on the order and meaning of the article in its entirety. The central text has undergone various changes to meet your requests:
.- the text has been rearranged to give more coherence to the epigraphs
.- A TGEU map has been added, which graphically illustrates what we are explaining ... at the current time and in our Europe
.- Some points have been incorporated into the text that gives it reflexivity (for example, the new resolution of the European Parliament)
It is done with the Word Tracking changes so that it is easy for you to follow
Point 4:
Conclusion: this section is too long and includes new information (on intersex)... cut the section down and use it to expand on the relevance of points not add new things. Put those ideas earlier
Response 4:
Thank you very much for your appreciation. We believe that you are right. We have shortened the conclusions. This shortening helps the reader to set ideas. The important affirmations that were made previously, have been distributed by the text
Point 5:
Moderate English changes required
Response 5:
To correct the language of the text, a revision of style was commissioned to the Proof-Reading Service company. The certificate is attached (please, see document). In addition, two collaborators, one native American English and the other British English, from our university department has been asked to review it again. Some modifications have been introduced

Reviewer 2 Report
It would be helpful if the list of abbreviations appeared at the beginning of the text, to aid comprehension.
The reference at your line 205 to GATE, requires amplification.
Your sentence at lines 124 to 126 requires clarification.
Your concluding sentence at lines 200-202 seems a non sequitur. The existence of a human right is not dependent on its uniform observance. Jurisdictions that fail to observe the human rights are in breach of their human rights duties, but do not show that the human right is not real. That is, countries in violation fail to make the human right a reality.
You might consider "Management of Intersex Newborns: Legal and Ethical Developments" by B.M. Dickens, in International Journal of Gynecology and Obstetrics (2018) vol. 143, pages 255-59.
Author Response
Point 1:
It would be helpful if the list of abbreviations appeared at the beginning of the text, to aid comprehension
Response 1:
Thank you very much for your support. It is already set. It is true that it is clearer to have it at the beginning
Point 2:
The reference at your line 205 to GATE, requires amplification
Response 2:
Thank you. We totally agree with you. It is already modified and defined
Point 3:
Your sentence at lines 124 to 126 requires clarification
Response 3:
Thank you very much because you have made it reflect on the order and meaning of the article in its entirety.
It is not only those lines to modify, we believe that the text needed a complete reordering to make it
more sense. We have also incorporated some elements that we believe facilitate understanding and reading.
Changes have been made with the Word Tracking changes so that it is easy for you to follow
Point 4:
Your concluding sentence at lines 200-202 seems a non sequitur. The existence of a human right is not dependent on its uniform observance. Jurisdictions that fail to observe the human rights are in breach of their human rights duties, but do not show that the human right is not real. That is, countries in violation fail to make the human right a reality
Response 4:
Thank you very much for your support. It is difficult to transfer ideas so endowed with sensitivity and strength to a different idiomatic construction (we talk, live and think in Spanish). We completely agree with what you say. In civilized countries it cannot be otherwise, we never wanted to express the opposite. In the team of researchers, we have decided to remove that phrase you mention and rewrite the conclusions. We believe that they are now clearer and more concise
Point 5:
You might consider "Management of Intersex Newborns: Legal and Ethical Developments" by B.M. Dickens, in International Journal of Gynecology and Obstetrics (2018) vol. 143, pages 255-59
Response 5:
Thank you very much for inviting us to read this interesting work. In fact, I have offered it to a 'key informant', that is, the mother of an intersexual child. For her it is important, and she has offered her collaboration to help us spread that knowledge
Point 6:
Moderate English changes required
Response 6:
To correct the language of the text, a revision of style was commissioned to the Proof-Reading Service company. The certificate is attached (please, see document). In addition, two collaborators, one native American English and the other British English, from our university department has been asked to review it again. Some modifications have been introduced
